# Microbiomes of Primary Soils and Mining Heaps of Polymetallic Ore Quarries

Ekaterina Dorogaya [1,*], Evgeny Abakumov [2,*], Aleksei Zverev [2], Evgenia Novikova [2], Mikhail Garshin [1], Aleksandr Minnegaliev [3] and Ruslan Suleymanov [1,4]

[1] Ufa Institute of Biology, Ufa Federal Research Centre, Russian Academy of Sciences, 450054 Ufa, Russia; garshin.mixail@yandex.ru (M.G.); soils@mail.ru (R.S.)

[2] Department of Applied Ecology, Saint-Petersburg State University, 199034 Saint Petersburg, Russia; euhennovi@gmail.com (E.N.)

[3] Department of Geology, Hydrometeorology and Geoecology, Ufa University of Science and Technology, 450076 Ufa, Russia; minnegaliev.aleksandr@rambler.ru

[4] Department of Geodesy, Cartography and Geographic Information Systems, Ufa University of Science and Technology, 450076 Ufa, Russia

[*] Correspondence: ekaterina.s.dorogaya@gmail.com (E.D.); e.abakumov@spbu.ru or e_abakumov@mail.ru (E.A.); Tel.: +7-9111-969-395 (E.A.)

**Abstract:** This research evaluates the development of microbiomes in primary soils, forming in various mining dumps in the arid terrain in the Republic of Bashkortostan, Russia. A metagenomic analysis of the communities was performed by sequencing extended gene sequences. The evaluation of the agro-chemical properties was in accordance with conventional pedology methods. Inverse voltammetry was used to measure the heavy metals (lead, cadmium, mercury, zinc, copper, and nickel) and arsenic content. In all the samples studied, Actinobacteria and Proteobacteria phylas dominated, and, in smaller numbers, Acidobacteria and Bacteroidetes were present. In the natural samples, the proportion of Actinobacteria was higher, and the proportions of Proteobacteria and Bacteroidetes were lower than in the samples from anthropogenically disrupted soils. Verrucomicrobia bacteria and Thaumarchaeota archaea were not found in the forming soils of the Kulyurtau and Tubinsky quarries, although in all other samples, there was a significant content of representatives of these types. Soil formation was observed at the Kulyurtau and Tubinsky mines, with a self-restoration period of more than 30 years. The microbial communities of the forming soils were similar in species richness to the background soils, and the alpha diversity showed a high level of dispersion, although the beta diversity had a different clustering, but the absence of Verrucomicrobia and Thaumarchaeota phyla in the samples from both sites indicates the underdevelopment of new soils compared with the natural background. Agrochemical indicators showed a dependence on the type of growing vegetation and the degree of anthropogenic load, and the correlation with the microbial composition of soils was traced poorly.

**Keywords:** soil degradation; quarries mining; DNA sequencing; microbial communities; soil self-healing

## 1. Introduction

Mineral extraction is crucial in obtaining resources for the industry, energy, and construction sectors and many other areas of human activity. Open-pit mining is one of the widespread and most economically profitable methods of extracting minerals [1]. Yet, it leads to the local degradation and transformation of the ecological environment. Open-cast mining significantly changes such important components of the environment as soils, surface and ground waters, and plant and animal communities [2–5]; a negative impact on the air quality and pollution of the adjacent territory with heavy metals is observed [6,7]. Still, the parameters of the disrupted ecosystem are restored for decades and, as a result,

usually differ from the background ecosystems of the area that existed before the beginning of open-pit mining [8–10].

Most research activities in assessing the restoration of the natural environment in the zones of open-cut mines focuses mainly on vegetation communities. This approach is applied as the main ecological criterion of reclamation quality since the restoration of vegetation leads to the renewal of ecosystem functioning [9–11].

Yet, the necessary condition for stable ecosystems is soil restoration in the disrupted areas [12]. Open-cut mining leads to significant changes in the initial physical, chemical, biochemical, and biological properties of the soil cover in the surrounding areas, with the soil being completely destroyed while mining.

The removal of the soil profile to the parent rock, as well as a disturbance of the vegetation and soil cover of nearby areas, significantly changes the ecosystem of the region [13]. The natural balance in the cycle of substances changes, chemical elements are involved in active geochemical migration, and the biomass of plants and microorganisms is redistributed [6]. Negative consequences such as a reduction in soil moisture and nutrient removal [14,15], heavy metal pollution [16,17], the weathering of rocks, and a change in the pH of the environment [18] are observed. Nearby areas are contaminated by sub-dump effluents from mining operations by infiltration and particulate erosion material by aerogenic means [19]. Soil movement and the creation of pit dumps lead to the removal of rocks with an increased content of toxic substances to the surface. This makes quarries less suitable for vegetation development, especially in arid and semi-arid areas where self-regeneration is slow due to environmental factors [8,20].

Thus, soil formation is limiting in the restoration of quarry ecosystems. Soil microorganisms play a key role at the initial stage of soil formation. The accumulation and transformation of organic matter and nitrogen, and hence fertility, are directly connected with their activity. The ecosystem function of the soil is also supported by microorganisms responsible for the balance of nutrition elements and capable of rapidly adapting to environmental changes, supporting the sustainable development of the vegetation cover [21].

The content of the microbial biomass in soils is characterized not only by its quantitative composition but also by a specific set of microbial communities with a certain sequence of distribution in the profile for each soil type [21,22]. The composition of microbial communities and their abundance are self-regulating parameters directly dependent on the properties of the habitat and nutrient substrate. Soil characteristics such as the pH; the organic carbon and nutrient content; moisture saturation and the frequency of moisture and drying periods; the granulometric composition and porosity; the presence of heavy metals; and other physical, chemical, and mechanical properties of soils directly affect the species composition of bacterial communities, determining the ways of their nutrition and ability to maintain their existence under given conditions [23–25]. Favorable conditions for the development of microorganisms contribute to the accumulation of their biomass and increase species diversity. And vice versa: in the absence of conditions for maintaining the stable vital activity of certain types of microorganisms, their development slows down, and their composition and quantity degrade. Such communities are gradually replaced by species better adapted to new conditions [26,27].

At the same time, the presence of a large diversity of species of microbial communities in soils, their significant biomass, and the ability to self-regulate largely determine the development of soils and their resistance to negative external influences [28]. Thus, for technogenically disturbed areas, the microbial composition serves as an indicator of the degree of soil recovery and indicates the nature of internal processes, being one of the most sensitive ecological indicators of different stages of pedogenesis. In combination with agrochemical predictors, the composition of microorganisms can serve as a criterion for assessing the suitability of restored soils for a particular type of economic activity [22,29,30].

The given study investigates the soil microbiomes in natural self-healing in quarries and polymetallic ore dumps. Samples of anthropogenically disturbed and natural background soils were studied by metagenomic methods. To analyze the conditions of soil

formation, the relationship between the composition of the microbial community and the material properties of the quarry dumps (agrochemical, morphological, heavy metal, and arsenic content) were evaluated. This study was carried out in the Bashkir Trans-Urals, Russia, in 2021.

## 2. Materials and Methods

The Republic of Bashkortostan, located at the boundary of the East European (Russian) Plain and the southern part of the mountainous Urals, is one of the developed centers of the mining industry in the Russian Federation (Figure 1).

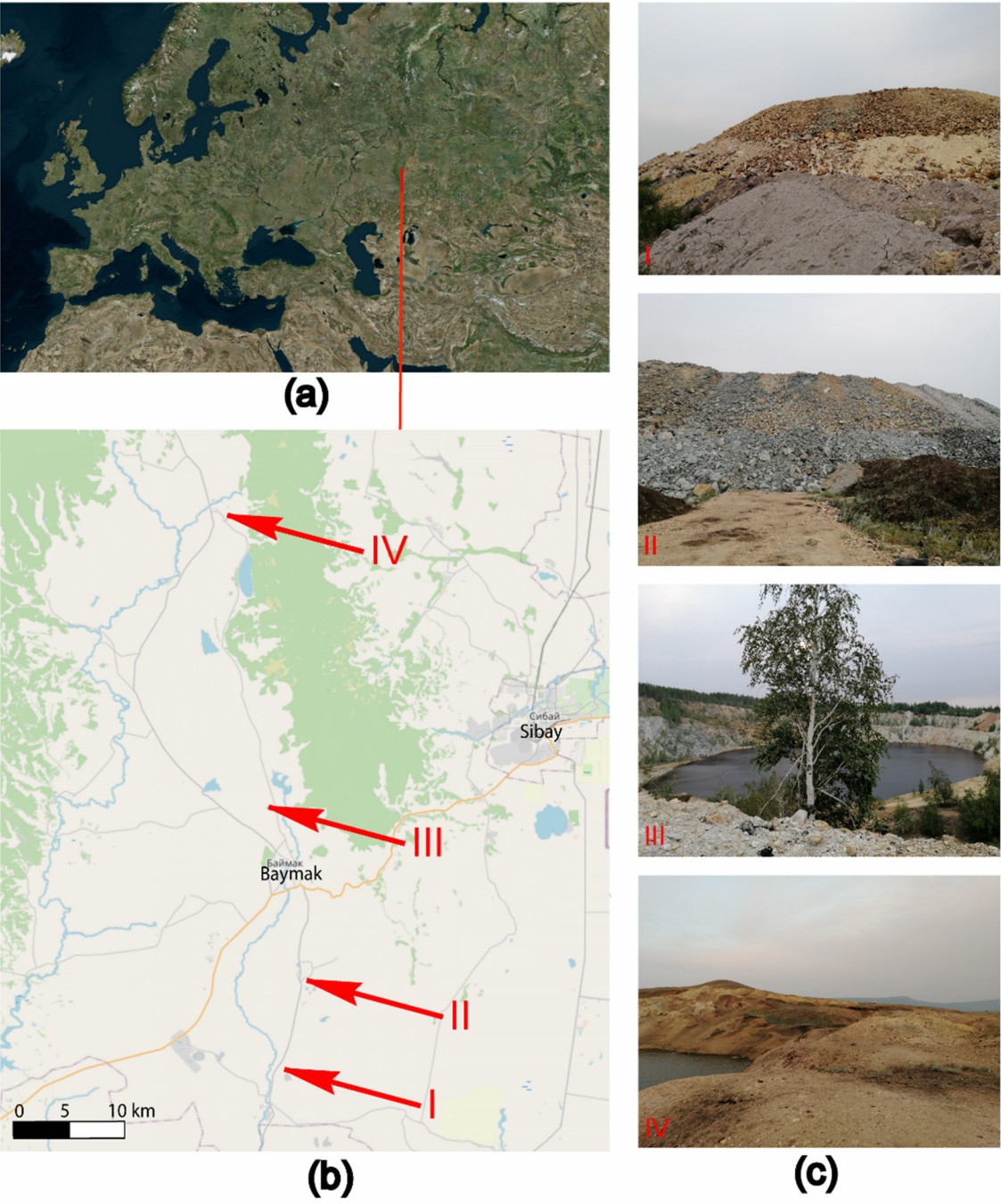

**Figure 1.** Work areas (**a**,**b**) and location of sample plots (**c**). Quarries and overburden dumps of mines: I—Tuba-Kain (Section 1-2021), II—Semenovsky (Section 3-2021), III—Kulyurtau (Section 5-2021), and IV—Tubinsky (Section 7-2021).

There are more than three thousand mineral deposits, including oil, natural gas, brown and hard coal, rock salt, limestone, gypsum, peat, and sapropel, as well as various metals, polymetallic ores, and decorative stones [31–33]. The unique accumulation of large deposits of copper-coal ores, gold placers, bauxite, zeolites, and many other minerals traditionally extracted by open-pit mining has led to the emergence of a large number of technogenically disturbed territories, including the largest quarries near the cities of Sibai and Uchaly [34–37]. The study of such territories and the development of reclamation methods is of vital importance for the Republic of Bashkortostan. At the same time, many works aimed at these topics focus on the physicochemical and sanitary indicators of soil [38,39], and the microbial composition is practically not studied [29].

The following dumps of deposits located in the south-east of the Republic of Bashkortostan in the Baymaksky district were investigated: Tuba-Kainskoe (1-2021)—refers to gold-barite-bearing oxidized gold ores; Semenovskoe (3-2021)—sulfur-barite gold-bearing with the inclusion of oxidized gold-bearing gold ores [40]; and Kulur-Tau (5-2021) and Tubinsky (7-2021)—mines that were used for the extraction of oxidized gold-bearing ores and placer gold, and now their development is completed, and the pits are flooded (Figure 1).

Overburden rocks are deposited during mining by forming dumps, the chemical composition of which, according to literature data [6,41], is close to the composition of mined ores and contains significant amounts of the main mined metals—copper, zinc, silver, and gold.

In accordance with the soil and climatic zoning of the territory of the Republic of Bashkortostan [42], the study area belongs to the South Trans-Ural low steppe and is located within the Sakmaro-Tanalyk high plain in the zone of the landscapes of the leeward plains' steppe barrier shadow. The average absolute height of the land surface varies from a 400–450 m Baltic height system (m BS) in the southern part to a 500–600 m BS in the northern part. The soils contain dense, rubbly podzolized chernozems and ordinary chernozems of clayey and heavy loamy texture. Alluvial–diluvial sediments act as soil-forming rocks. Soil acidity varies from neutral and alkaline to close to neutral when moving from south to north. By the nature of moisture, the studied region is dry with an insufficient moisture supply during the growing season (Table 1).

**Table 1.** Climatic characteristics of the studied cites [42].

| Factor | Value |
| --- | --- |
| Humus horizon deepness | 10–50 cm |
| Bulk humus content | 4–6% |
| Sunshine period | 1950–2000 h year$^{-1}$ |
| Average air temperature | 1.5–2 °C |
| Average July air temperature | 17.5–18 °C |
| Average precipitation | 350–400 mm (mm) |
| Amount of precipitation during the warm period | 250–300 mm |
| Humidity coefficient | <0.4 |
| Evaporation | 583 mm year$^{-1}$ |
| Average number of days with atmospheric drought | 40–45 days year$^{-1}$ |
| Average wind speed | 3.5–4 m s$^{-1}$ (m c$^{-1}$) |

Vegetation is represented by West Siberian and North Kazakhstan types on herb–grass–soddy–grass steppes in the southern part and arable lands and steppe meadows on the site of pine reed-bramble forests with secondary birch forests in the central and northern part of the study area. The area includes the Baymak-Buribay ore district (copper, gold, zinc, lead, nickel, cobalt, and manganese mining) and the Baymak ore district (gold mining) and has a high anthropogenic load on the natural environment due to the mining industry and intensive agriculture. The presence of a developed complex of mining enterprises led to the emergence of extensive areas of technogenically disrupted soils of various ages and degrees of self-restoration.

In order to study the primary soil-forming process, samples of the upper soil horizon (0–20 cm (cm)) were taken from quarries and overburden rock dumps in 4 different locations. Each sample of a technogenically disrupted area corresponded to a soil sample from a nearby background area with an undisturbed soil surface. The background was chosen taking into account the minimization of the possible transfer of pollutants, particularly in places with a higher hypsometric level (Table 2 and Figure 1).

**Table 2.** Description of the studied areas and selected samples.

| Sample No. | Plot | Soil Cover | Vegetation Cover | Soil Age, Years | Sample Description |
|---|---|---|---|---|---|
| 1-2021 | Tuba-Kain (heap) | Missing | Missing | 5–7 | Dry; mottled light gray, almost white, with inclusions of white, red, and dark gray; coarse-grained sandy dense, without plant remains |
| 2-2021 | Tuba-Kain (background) | Continuous Leptosols, up to 11 cm | Continuous steppe vegetation | | Dry; dark gray, relatively nonhomogenous in color, with a yellowish tint; sandy clumpy; lots of plant roots |
| 3-2021 | Semenovsky (heap) | Missing | Locally vegetated by grass spots | 1–2 | Dry; mottled light gray, with inclusions of white and yellow (gravel and rubble); crumbly; a small number of inclusions of vegetation roots and plant debris |
| 4-2021 | Semenovsky (background) | Continuous Leptosols, up to 17 cm | Continuous steppe vegetation | | Dry; uniformly dark gray, almost black, with a yellowish-reddish hue; sandy crumbly fine and medium crumbly; large amounts of plant roots and plant debris |
| 5-2021 | Kulyurtau (quarry) | Localized newly formed soil spots | Young birch forest stands | >30 | Dry; mottled dark gray (soil part) with inclusions of white and yellow (wood and rubble parts); soil fraction powdery sandy; a large number of inclusions of roots and plant residues |
| 6-2021 | Kulyurtau (background) | Continuous Leptosols, up to 10 cm | Continuous steppe vegetation | | Dry; dark gray, almost black, with a yellowish tint; coarse- to medium-compound sandy crumbly; many inclusions of roots and plant debris |
| 7-2021 | Tubinsky (quarry) | Localized newly formed soil spots | Partially degraded steppe vegetation cover | >60 | Dry; mottled light gray, with a reddish tint and with inclusions of white and red; sandy, and the structure is not expressed (the sample is a mixture of sandy and sandy material); a small number of inclusions of plants remain |
| 8-2021 | Tubinsky (background) | Continuous Leptosols, up to 8 cm | Continuous steppe vegetation | | Dry; uniformly dark gray, almost black; powdery sandy; many inclusions of coarse gravel and rubble, and roots and plant debris |

The age of the quarries and pit dumps, and the approximate start of the initial soil formation were determined based on the analysis of multi-temporal satellite images (the Google Earth platform was used).

A sample of about 1 kg of fine-grained soil was taken from each area, which was then separated by quarting and used for agrochemical and microbiological analyses and an analysis for the content of heavy metals (lead (Pb), cadmium (Cd), mercury (Hg), zinc (Zn), copper (Cu), and nickel (Ni)) and arsenic (As). Samples for determining the taxonomic composition of the microbiome were stored in a freezer at $-20$ degrees Celsius ($^\circ$C) until the study, and the rest of the sample was air-dried. Samples for each of the analyses were prepared according to the methodology of the experiment.

The agrochemical studies were performed in accordance with methods generally accepted in soil science: the determination of the total humus content was estimated according to a method by Tyurin and was terminated by that of Orlov and Grindel [43], the values of soil acidity in water extraction (pH $H_2O$) were determined by a potentiometric analysis, and alkaline-hydrolyzed nitrogen was established by the Kornfield method [44,45].

An analysis of the bulk content of heavy metals (Pb, Cd, Hg, As, Zn, Cu, and Ni) and the labile forms of heavy metals (Zn, Cu, and Ni) and As in the soil were performed by inversion voltammetry [46]. The experiments were carried out according to MU 31-11/05 (FR.1.31.2005.02119) and MU 31-18/06 (FR.1.31.2007.03301) [47]. The values of the maximum permissible concentrations (MPCs) and approximate permissible concentrations (APCs) of Hygienic Standard 1.2.3685-21 from 28 January 2021 [48] for evaluating the bulk content of heavy metals and their labile forms were used.

DNA extraction was performed using the MN FastDNA Spin Kit (Macherey-Nagel, Düren, Nordrhein-Westfalen, Germany) for soil total DNA extraction. At the stage of the mechanical destruction of the sample, a Precellus 24 homogenizer (Bertin, Montigny-le-Bretonneux, France) was additionally used. Quality control of the isolated DNA was performed by a visual evaluation of electrophoregrams in 1 percent (%) agarose gel, as well as by control PCR using sequencing primers. Sequencing was performed on an Illumina MiSEQ sequencer (Illumina, San Diego, CA, USA), according to the manufacturer's protocol. The primers used were F515 (GTGCCAGCMGCCGCGGCGGTAA) and R806 (GGACTACVSGGGTATCTAAT), which allow for the sequencing of the V4 hypervariable region [49]. Initial reads were deposited for further inclusion in publications. Sequences were processed using the R language (v3.6.3). The analysis was based on the use of the dada2 (v1.14.1) [50] and phyloseq (v1.30.0) packages [51]. The main steps were (i) quality filtering and merging (the filters were forward read length of >240, reverse read length of >160; discard reads that match against the phiX genome; per base quality of >2; maximal expected error = 2; Ns are not allowed); (ii) de novo ASV picking; (iii) the removing of chimeras; (iv) taxonomic annotation; and (v) reference tree construction. As a training data set for the taxonomic annotation, the SILVA (Quast et al., 2013) (v138) reference database was used. During the analysis, sequences were filtered for the reading quality, amplicon sequence variants (ASVs) were identified using machine learning algorithms, and forward and backward sequences were combined. An ASV representation table was generated. Taxonomic annotation was performed using a I Bayesian classifier, and the SILVA library of reference reads was used as a training set of sequences (124) [52].

### 3. Results

The conducted studies revealed that soil-like bodies of disrupted areas are at the stage of the primary soil-forming process and feature a weakly formed profile (Kulyurtau and Tubinsky open-pit dumps) or its complete absence (Tuba-Kain and Semenovsky mine dumps). Erosion processes include weakly pronounced water erosion (mainly during snowmelt) and wind erosion, which is typical for regions with an arid climate. During a field inspection of sites on the steep slopes of dumps and quarry walls, traces of planar washouts and water erosion in the form of a brittle cemented crust of a small thickness, cut by cracks and runoff furrows (Kulyurtau and Tuba-Kain dumps), were revealed. Wind erosion was

more pronounced, and signs of aeolian transport were found on all anthropogenic-disturbed plots in the form of deposits in the surface irregularities of fine fractional material with a characteristically layered structure. Soil formation and its elementary components (litter accumulation, humus formation) were observed on separate local plots not subject to erosion and mechanical transport of material down the slope. The Kulyurtau and Tubinsky open-pit dumps show the presence of primary soils: at the Kulyurtau site, an accumulation of medium-thickness litter under the canopy of young birch trees was present; at the Tubinsky site, turf was formed within the bottom of the pit, and the formation of a weakly expressed abscission was traced. At younger sites of the quarries and dumps, as well as at the steeply sloping areas of the Kulyurtau and Tubinsky mines, an accumulation of litter and a formation of a humus-accumulative horizon were not present.

### 3.1. Soil Agrochemical Characteristics

The fine-grained soil sampled from the quarries and overburden rock dumps was mainly poor in organic matter. An exception to this was a sample taken from the dumps Kulyurtau quarry (5-2021), with 6.1% of total humus, which corresponds to a high content and exceeds similar readings in the background area of this quarry (6-2021), with a value of 4.5% of total humus [53].

The content of total humus in the background plots was characterized by essential heterogeneity. The background soils in the area of the Tuba-Kain mine with a low content of total humus (2.8%) had the lowest rate, and the background soils of the Tubinsky quarry with high content values (9.6%) showed the highest supply (Table 3).

**Table 3.** Soil agrochemical parameters.

| Sample No. | Plot | pH $H_2O$ | TOM, % | Alkaline Hydrolizable Nitrogen, Milligram Kilogram$^{-1}$ (mg kg$^{-1}$) |
|---|---|---|---|---|
| 1-2021 | Tuba-Kain (heap) | 7.35 | 0.3 | 0 |
| 2-2021 | Tuba-Kain (background) | 6.68 | 2.8 | 126 |
| 3-2021 | Semenovsky (heap) | 7.27 | 2.1 | 98 |
| 4-2021 | Semenovsky (background) | 6.57 | 3.8 | 182 |
| 5-2021 | Kulyurtau (quarry) | 6.16 | 6.1 | 336 |
| 6-2021 | Kulyurtau (background) | 6.63 | 4.5 | 224 |
| 7-2021 | Tubinsky (quarry) | 7.62 | 1.4 | 84 |
| 8-2021 | Tubinsky (background) | 5.77 | 9.6 | 448 |

The content of alkaline-hydrolyzable nitrogen in the soils, a determinant of soil fertility, directly correlates with the amount of organic matter in the samples studied. The maximum amount of alkaline-hydrolyzable nitrogen among all the samples was found in the area of background soils with a high content of total humus 8-2021 and corresponds to the indicators of high security (448 mg/kg). A complete absence of alkaline-hydrolyzable nitrogen was found in the sample from the Tuba-Kain dump site (1-2021), where the content of humus reached very low, almost to the level of error, values (0.3%). The provision of other sites was high and ranged from 84 (7-2021) to 336 (5-2021) mg/kg of soil. The highest provision of alkaline-hydrolyzable nitrogen among the samples of technogenically disturbed soils was revealed in the Kulyurtau open pit (Table 3). The only exception to this was observed at the site of the Tubinsky quarry (7-2021), where the reaction of the environment was slightly alkaline—7.62. In general, the pits' pH $H_2O$ had higher values, i.e., a more alkaline environment, than the pH $H_2O$ of the corresponding background sites everywhere, except for at Kulyurtau, where the agrochemical indicators of primary soils on the abandoned pit dumps were generally better than the indicators of undisturbed

background soils (Table 3). The pH value of the studied samples from different sites corresponded to the neutral pH values typical for the soils of this region and ranged from 5.77 (8-2021) to 7.35 (1-2021). Increased pH $H_2O$ values are not characteristic of anthropogenically disturbed soils, but due to the strong dependence of environmental acidity on the chemical composition of soils and the initial pH $H_2O$ value, as well as natural-climatic conditions in the study area, it is quite difficult to derive any dependence between these factors.

### 3.2. The Trace Elements (Heavy Metals) and Arsenic Content

An analysis of the content of heavy metals such as Pb, Cd, Hg, Zn, Cu, and Ni in bulk concentration and Zn, Cu, and Ni in mobile form, as well as an analysis of the arsenic concentration in the quarry material, showed that exceedances of the maximum permissible concentration (MPC) and the approximate permissible concentration (APC) were observed only in the background area near the Tubinsky quarry, where the bulk concentration of lead, by 1.6 times, copper, by 1.3 times, and cadmium, by 1.75 times, exceeds the established APC, and the content of zinc in labile form exceeds the MPC by 1.4 times (Table 4).

**Table 4.** The content of trace elements and arsenic and soils.

| Samp. No. | Bulk Content, mg kg$^{-1}$ | | | | | | | Available Form, mg kg$^{-1}$ | | |
|---|---|---|---|---|---|---|---|---|---|---|
| | **Pb** | **Cd** | **Hg** | **As** | **Zn** | **Cu** | **Ni** | **Zn** | **Cu** | **Ni** |
| 1-2021 | 18 | 0.16 | <0.1 [a] | 1.3 | 37 | 26 | 34 | 3.8 | 0.78 | 1.5 |
| 2-2021 | 25 | 0.23 | <0.1 [a] | 1.6 | 56 | 33 | 41 | 6.1 | 1.1 | 1.9 |
| 3-2021 | 16 | 0.26 | <0.1 [a] | 1.5 | 32 | 21 | 30 | 4.1 | 0.69 | 0.94 |
| 4-2021 | 24 | 0.31 | <0.1 [a] | 1.3 | 48 | 27 | 38 | 5.1 | 0.74 | 1.3 |
| 5-2021 | 19 | 0.22 | <0.1 [a] | 1.4 | 35 | 25 | 31 | 4.6 | 0.83 | 1.1 |
| 6-2021 | 16 | 0.34 | <0.1 [a] | 1.2 | 34 | 28 | 26 | 3.6 | 0.92 | 0.83 |
| 7-2021 | 21 | 0.31 | <0.1 [a] | 1.1 | 63 | 37 | 35 | 7.8 | 1.4 | 1.3 |
| 8-2021 | 214 | **3.5** | <0.1 [a] | 2.4 | 98 | **166** | 21 | **32** | 2.9 | 0.53 |
| MPC | 130 | 2 | 2.1 | 10 | 220 | 132 | 80 | 23 | 3 | 4 |
| APC | APC | APC | MPC | APC | APC | APC | APC | MPC | MPC | MPC |

[a] is the lower limit of the method definition; the values in bold are those exceeding the sanitary norms. MPC—maximum permissible concentration; APC—approximate permissible concentration.

The copper content in labile form was close to the MPC values (2.9 mg kg$^{-1}$, while the norm is 3 mg kg$^{-1}$ of copper in soil), and the content of total zinc and arsenic were the highest among the studied samples but did not exceed sanitary norms. At other sites, for both the background and anthropogenically disrupted sites, the content of heavy metals and arsenic did not exceed the MPCs and APCs. High values of the measured parameters of the background undisturbed soil plot 8-2021 must be connected with the general ubiquitous natural increase in the heavy metal content in the studied region [54,55] and the significant content of humus (9.6%) in this plot, which accumulates harmful substances, contributing to their accumulation in the soil.

### 3.3. Soil Microbiomes

The sequencing yield was 832,583 raw reads in total. After filtering, mering, and grouping into ASVs, the total amount was 389,526. In the downstream analysis, 28 samples with a mean of 13,911 reads per sample were included. For an alpha-diversity calculation, the data were rarefied to the minimal depth of 9684 sequences per sample.

The taxonomic composition of the studied samples was mainly represented by the sum of the following prokaryotic phyla: Acidobacteria, Actinobacteria, Bacteroidetes, Chloroflexi, Cyanobacteria, Firmicutes, Gemmatimonadetes, Patescibacteria, Planctomycetes, Proteobacteria and Verrucomicrobia, and Thaumarchaeota archaea (Figure 2).

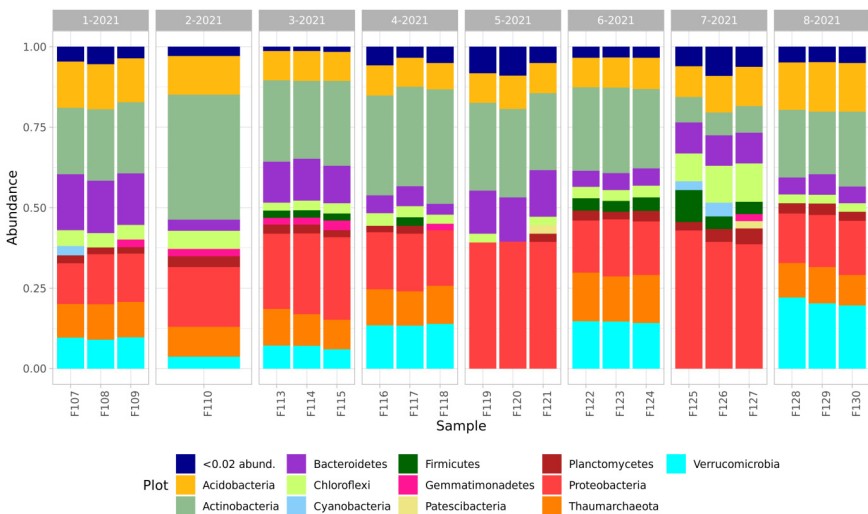

**Figure 2.** Taxonomic composition of soils of the studied sites.

The taxonomic composition of all the samples was dominated by Actinobacteria and Proteobacteria phyla, and Acidobacteria and Bacteroidetes were present in smaller amounts. In the samples from the quarry dumps compared with those of the background soils, there was a decrease in the composition of the microbial communities of Actinobacteria and an increase in the proportion of Proteobacteria and Bacteroidetes, and a significant difference in the composition of Acidobacteria between the comparison samples was not revealed. In all the soil samples of the natural background and the samples from the quarries and the overburden dumps of the Tuba-Kain and Semenovsky mines, there was a significant number of representatives of phylum Verrucomicrobia and Thaumarchaeota. At the same time, the number of these phylum in the forming primary soils in the dumps of the Kulyurtau and Tubinsky quarries was extremely small and did not exceed 2%.

The alpha diversity of the microbial communities was estimated by ASV and the Simpson index (Figure 3).

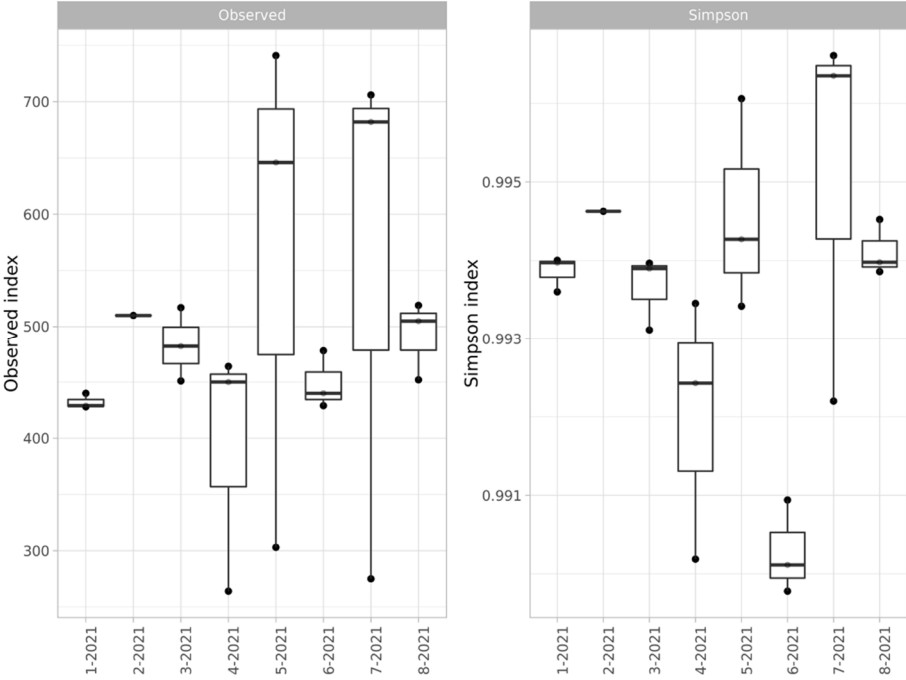

**Figure 3.** Alpha diversity (ASV representation and Simpson index) of bacterial communities of soils of the studied sites.

The alpha diversity showed an unusually high level of dispersion for the samples from the Kulyurtau and Tubinsky open-pit dumps (5- and 7-2021, respectively). This effect may be related to the partial overlapping of the waste rock areas with areas of forming soil or areas of pasture digression. The presence of such a scatter does not allow for a direct comparison of the levels of the diversity of dump soils and background soils; nevertheless, it can be concluded that the species richness is similar.

The beta biodiversity (Figure 4) suggests the presence of a pronounced clustering of the dump soils, although the microbial community of the Semenovsky dump (3-2021) was similar to the group of background soils.

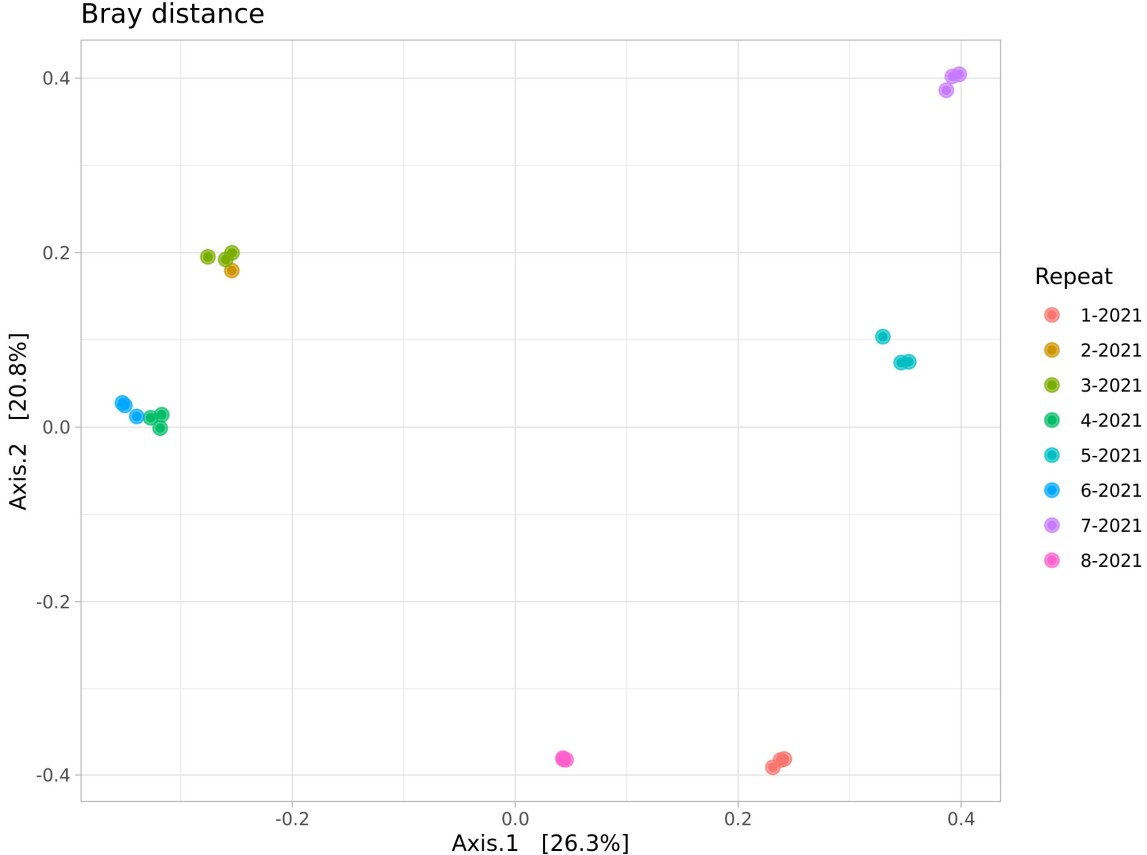

**Figure 4.** PCoA ordination of beta diversity (Bray's distance) of microbial communities of soils of the studied sites.

This peculiarity is clearly visible on the graphs of the relative representation of microbial phyla: the samples of the background soil and the dump soil from Semenovsky (3-2021 and 4-2021) differ in the presence of a greater proportion of Bacteroidetes and the presence of Gemmatimonadales representatives in the latter. It is also worth noting that there is some distance between the clusters of the background soils: the first is represented by soil communities near the Kulurtau (6-2021), Tuba-Kain (2-2021), and Semenovsky (4-2021) quarries and the second by the community of background soils of the Tubinsky quarry (8-2021).

## 4. Discussion

Signs of the formation of primary soils were observed on the overburden dumps of the Kulyurtau and Tubinsky quarries, the period of self-regeneration of which is more than 30 years and more than 60 years, respectively. In both cases, the process takes place in areas that are little exposed to erosion impacts and occupied by woody (Kulurtau) or herbaceous (Tubinsky) vegetation. No processes of litter accumulation and the formation of primary soils were detected at the younger quarries of the Tuba-Kain and Semenovsky mines.

### 4.1. Soil Agrochemical Characteristics

The direction and intensity of soil formation is directly related to biological parameters. Microbial communities, in turn, show a strong dependence on both external environmental conditions and the properties of the substrate in which their life activity takes place. In anthropogenically disrupted areas, these conditions change significantly compared to natural conditions. The following agrochemical characteristics were used to assess the properties of the studied soils: the pH in water extracts, the total organic matter content (humus), and the amount of alkaline-hydrolyzed nitrogen.

Organic matter and nitrogen in soils significantly affect the content of the total microbial biomass, providing nutrition to microorganisms. In addition, organic matter and nitrogen content are the leading indicators of soil fertility [56,57]. The level of acidity of the medium pH has a significant influence on the microbial composition of soils. As was shown in the work by Lauber et al. [58], the effect of the soil pH $H_2O$ on the composition of bacterial communities is evident even at a relatively low level of taxonomic resolution.

It was found that, according to the agrochemical characteristics, the areas of open pits and overburden dumps have a lower content of total humus and alkaline-hydrolyzable nitrogen and higher pH values, reaching slightly alkaline values compared to the background areas. An exception to this is the open-pit Kulyurtau, where the content of total humus and the alkaline-hydrolyzed nitrogen in the samples from the dumping site were higher than in the samples from the background (6.1% and 4.5% of humus, and 336 mg kg$^{-1}$ and 224 mg kg$^{-1}$ of alkaline-hydrolyzed nitrogen, respectively), and the pH $H_2O$ values were lower (pH $H_2O$ = 6.16 for the dumping site, and pH = 6.63 for the background). This is probably due to the presence of primary soils forming at the overburden dump site and the presence of different types of vegetation (woody at the dump and steppe at the background) at the compared sites, which gives an unequal contribution to the content of nutrients in the upper soil horizon [59].

### 4.2. The Trace Elements (Heavy Metals) and Arsenic Content

It is well known that the development and activity of microbial communities are strongly affected by the degree of anthropogenic pollution, including pollution with heavy metals [60,61]. An analysis of the content of some heavy metals in their bulk concentration (Pb, Cd, Hg, Zn, Cu, and Ni) and mobile form (Zn, Cu, and Ni) and an analysis of the concentration of arsenic were conducted to assess the degree of pollution of the studied soils (Table 4). An examination of the samples did not reveal exceeded sanitary standards at the open pits and overburden rock dumps, as well as at the comparison baseline sampling sites, except for a natural area near the Tubinsky open pit, where some heavy metals were above permitted levels. The exceeded values were detected for lead (bulk content 1.6 APC), copper (bulk content, 1.3 APC), cadmium (bulk content, 1.75 APC), and zinc (mobile form, 1.4 MPC), but no apparent effect of the exceeding sanitary norms of these heavy metals on the agrochemical and microbiological properties of the site was found.

### 4.3. Soil Microbiomes

The representatives of Actinobacteria and Proteobacteria were dominant in taxonomy composition, which is typical of most bacterial soil communities [62,63]. A notable proportion of Acidobacteria and Bacteroidetes was also found in each sample. The samples obtained from the primary soil formation sites in the Kulurtau and Tubinsky quarries were distinguished by a very small proportion of Verrucomicrobia and Thaumarchaeota phylum (less than 2%). Their content in the microbial community of the other samples varied for Verrucomicrobia from 5–6% (for the background soils of the Tuba-Kain pit dumps) to 20–22% (for the background soils of the Tubinsky pit) and from 10–15% of Thaumarchaeota. In spite of the poor study of Verrucomicrobia, these bacteria tend to dominate in many soil communities [64], and representatives of the archaea Thaumarchaeota are not only widely distributed in nature but also participate in nitrification, prevailing among ammonium-oxidizing soil microorganisms [65]. Consequently, the absence of such

important representatives of the microbial community in the forming soils of quarries indicates their underdevelopment and being at the initial stage of formation of a stable microbiome. Samples from the Kuliurtau and Tubinsky quarries had the highest number of minor types, the number of representatives of each of which did not exceed 2%. The dumps from the Tuba-Kain and Semenovsky quarries had a high similarity of microbial composition with the corresponding background soils, which was probably due to the relatively short age of abandonment since the end of mining at these sites (5–7 years and 1–2 years, respectively), during which the residual natural communities had not had time to degrade significantly. At the same time, an increase in the proportion of Bacteroidetes and a decrease in the number of Actinobacteria were observed in all the samples taken from areas with technogenically disturbed soils compared to the background. The high proportion of Actinobacteria phylum representatives in the samples is due to their wide distribution in the soils of arid regions with a neutral or alkaline environmental reaction [66]. However, it is also known that Actinobacteria play an important role in the decomposition of organic matter and take part in the carbon cycle [62]. Thus, when the amount of organic carbon decreases, their share in the microbial community decreases, which was observed for the samples of the studied quarry material. On the other hand, there was an increase in the proportion of representatives of Bacteroidetes, which also participate in the soil carbon cycle but thrive under more unfavorable environmental conditions, which is confirmed by the ubiquitous distribution of this phylum in all the studied ecosystems, especially in soils and human and animal intestines [67].

For the fine earth of the Tubinsky quarry, a significant increase in the number of Chloroflexi representatives, which are mixotrophs and facultative anaerobes fixing carbon as a result of a special 3-hydroxypropionate cycle, was also observed compared to the control. These microorganisms are confined to hydromorphic soils with a rather low content of organic matter and a high content of soluble iron [68]. Judging by the agrochemical analysis of this sample, the organic carbon content in it was one of the lowest among the studied soils (1.4%); in addition, the sampling site was located in a lowland within the flooded part of the quarry, which could probably promote the development of relevant microorganisms.

Representatives of the phylum Acidobacteria in the microbial community of all the studied samples had approximately similar amounts. Both the lowest and the highest proportions of these bacteria were observed in samples of natural undisturbed soils: the lowest at the background site of the Semenovsky pit dump (about 10–11%) and the highest at the background site of the Tubinsky pit ($\approx$20%). It is known that Acidobacteria are especially numerous in soil habitats with a low organic carbon content, and most of their representatives are considered as oligotrophic bacteria, with optimal development conditions under high environmental acidity [69]. On the contrary, the soils of the studied plots have a neutral or slightly alkaline environment and probably a sufficient amount of organic carbon in almost all the samples, so Acidobacteria were not dominant in the microbiome, although they were quite numerous. However, when comparing the agrochemical and sanitary characteristics of the samples, no correlation was found between changes in the soil characteristics and the composition of the Acidobacteria phylum.

An assessment of the alpha diversity of the microbial communities allowed us to speak about similar species richness between samples of quarry material and their background soils, which can probably be a consequence of an initial low level of microbial diversity, which is characteristic of poorly developed soils. According to the beta-diversity assessment, the fine earth of the quarries had a pronounced clustering of microbial communities, differing from the background soils to a greater or lesser extent.

Overall, two stages of evolution of technogenic soils were observed in this study, depending on the age of open-pit mining: an initial degradation of the disrupted natural environment for the areas of the overburdened mine dumps of Tuba-Kain (1-2021) and Semenovsky (3-2021) and a slow self-recovery of the abandoned mines at the areas of the Kulyurtau (5-2021) and Tubinsky (7-2021) open-pit dumps. In the initial two sites, the deterioration of the material properties of the dumps was well traced. At the same time,

the degradation at the Semenovsky plot, which is 1–2 years old, was much lower than that of the Tuba-Kain site, which was especially noticeable in the microbiome: the taxonomic composition and the alpha and beta diversity of the samples from the dump of the Semenovsky mine have a strong similarity with the natural background. In terms of agrochemical indicators, degradation proceeds faster, which was noticeable by a decrease in the content of the total humus and alkaline-hydrolyzable nitrogen. For the dump of the Tuba-Kain mine, which is 5–7 years old, an almost complete loss of these components was revealed, which actually indicates a loss of fertility, but from the side of the microbial composition, changes proceed significantly slower. The samples retained a similar taxonomic composition to the background natural site, and the alpha diversity decreased and a significant change in the clustering of the beta diversity was noticeable, but, in general, the microbial communities were more resistant to changes in external environmental conditions.

At the sites of the Kulyurtau and Tubinsky open pits, where no mining has taken place for many years, a direct correlation of the success of self-restoration with the duration of abandonment was not revealed. The dumps of both quarries had signs of primary soil formation, but the sample from the Kulyurtau quarry (more than 30 years of self-restoration), by the agrochemical characteristics, exceeds a sample from the Tubinsky quarry (more than 60 years of self-restoration). The microbial community data from both mines suggest a similar species richness to the background soils, although the beta diversity had a different clustering. The taxonomic composition of the microbial community of the Tubinsky open pit was more diverse than that of the Kulyurtau open pit, but the samples from both the first and second open pits contained almost no phylum of Verrucomicrobia bacteria and Thaumarchaeota archaea, which strongly distinguishes them from the natural background. Representatives of these phyla are usually contained in soils in appreciable quantities, from which we can conclude that even after 60 years of self-restoration in the quarry sites, the process of soil formation is still at the stage of development, and the microbial communities have significant differences from the communities of the original soils. At the same time, microbial adaptation to the changed environmental conditions occurs, which follows from the increasing proportion of microorganisms able to survive with less nutrients and a more alkaline pH. In addition, an increase in the total mass of microorganisms, the content of which did not exceed 2%, indicates the ongoing competition between species for reduced food sources.

Summarizing the results obtained, we can say that the process of the self-reclamation of technogenically disturbed soils observed in this study has no direct dependence on the initial agrochemical properties of soils and the content of heavy metals in them. The degradation of soil properties of disturbed areas proceeds with a fairly rapid loss of nutrients and a slower restructuring of microbiological communities. The primary soils, formed on the sites of abandoned quarries, show an inverse relationship: the increase in nutrient content proceeds faster than the recovery of microbiota and depends largely on the current anthropogenic load on the forming soils and the types of vegetation growing on these soils. These results do not contradict the general ideas about the development of microbial communities in disturbed soils [28] and the ranking of soil nutrient recovery rates and soil bacteria [70].

The obtained results clearly demonstrate the differences in the properties and composition, including the microbial composition, of the material of quarries and quarry dumps, depending on the duration of the self-restoration process and the level of anthropogenic load. In the future, such studies can be useful to assess the feasibility of applying certain methods of the soil restoration of technogenically disturbed territories, taking into account climatic conditions and specifics of soils of the Republic of Bashkortostan. Thus, a moderate loss of soil nutrient elements and a high similarity of the microbial composition of fine-grained materials from the Semenovsky pit dumps suggest that the method of restoration using the backfilling of local soils will have successful results for the reclamation of the disturbed area without significant costs for the relocation of a more fertile soil or a long time for self-restoration.

The dumps of the Tuba-Kain quarry, having a longer duration since the end of mining, were more affected by the degradation of agrochemical properties with changes in the microbiological composition of soils. In addition, local background soils also had a low level of bioproductivity. Consequently, the backfilling of local soils is not expected to be as successful as in the future for the Semenovsky pit dumps. In this case, additional restoration of soil nutrients and some erosion control measures would be required to reduce the leaching of ameliorants or fertile soils used for restoration. Probably, as in the case of both the Semenovsky and Tuba-Kain quarry dumps, the newly reclaimed soils may eventually have their own specificity and may be somewhat different from the local background soils. However, theoretically, the earlier the restoration activities are carried out, the lower the financial costs are required to be invested, not to mention a reduction in the environmental load on the area and the time of abandonment of technogenically disturbed territories.

In the case of quarries with signs of secondary soil formation, namely Tubinsky and Kuliurtau, probably the most effective and economically beneficial method of their recultivation will be their protection from further anthropogenic influences to intensify the process of self-restoration.

Thus, the assessment of the state of microbial communities in combination with agrochemical properties, the morphological features of the terrain, and factors of pollution and anthropogenic influence not only allow us to study the processes associated with secondary soil formation but also allow us to effectively assess the possibility of applying various methods of reclamation for technogenically disturbed areas.

**Author Contributions:** E.D.—field work, chemical soil analyses, data processing, and data curation; E.A.—conceptualization, resources, funding acquisition, and writing—review and editing; A.Z.—soil DNA extraction and bioinformatic data processing; E.N.—chemical soil analyses, statistical data treatment, and data visualization; M.G.—field research, data curation, writing—editing; A.M.—field and laboratory research, data treatment, and writing; R.S.—soil sampling, writing, figure preparation, and laboratory research. All authors have read and agreed to the published version of the manuscript.

**Funding:** This work was supported by the Federal budget of the Russian Federation, by a Grant to support for the creation and development of a World-Class Scientific Center Agrotechnologies for the Future (project no., 075-15-2022-322; date, 22 April 2022).

**Institutional Review Board Statement:** Not applicable.

**Informed Consent Statement:** Not applicable.

**Data Availability Statement:** The original contributions presented in the study are included in the article, further inquiries can be directed to the corresponding authors.

**Acknowledgments:** The authors thank the scientific park of the Saint-Petersburg State University for their help with analytical procedures (research centers: the Biobank, Chemical analyses, and Material research center and the Environmental safety observatory).

**Conflicts of Interest:** The authors declare no conflicts of interest. The funders had no role in the design of the study; in the collection, analyses, or interpretation of data; in the writing of the manuscript; or in the decision to publish the results.

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
