# Peer review of "Microbiomes of Primary Soils and Mining Heaps of Polymetallic Ore Quarries"

_applsci, doi:10.3390/app14083328_

Round 1

Reviewer 1 Report

Comments and Suggestions for Authors

The authors were trying study the microbial community compositions in various mining dumps in the arid terrain in the Republic of Bashkortostan, Russia. The study design and results were clearly presented, but the method and discussion sections need to be improved prior to publication. Please see my detailed comments below:

Line 34: one point that is not clear in the introduction is why you think study microbial community composition is important? there may be functional redundancy, and different microbes may perform the same or similar function, so function diversity is more important than community composition.

line 115: what does this table legend mean? need to update the table legend

Line 118: is it possible  to mark sample number in Figure 1 to better help your readers to identify the sample location

Line 143: separate the DNA extraction and sequence method and other physiochemical methods into different paragraphs.

Line 151: need more details about sequence processing, such as Phred score , reads with which length were removed, and adaptor contaminants?

Line 157: which version of SILVA was used?

Line 158: would suggest adding another section in the results part to show the sequence depth, before and after quality control sequence reads number, etc.

Line 228: general comment, be consistent about the text Font and Font size

Line 253: did you see any pattern of community composition at genus or family level?

Line 260-263: suggest move this to discussion and can have a more detailed discussion if it's important. The same for other parts, try not to reference previous studies in your results section as you have a separate discussion section.

Line 366: the discussion section needs to be improved. I did not see any previous studies that were discused against your results. in addition, seems like you observed difference among different sampling locations in community composition, diversity. but the discussion is not enough here to explain why and what does that mean.

Author Response

Dear reviewer!

Thank you for your comments, point by point replies are given below (tracked in yellow)

The authors were trying study the microbial community compositions in various mining dumps in the arid terrain in the Republic of Bashkortostan, Russia. The study design and results were clearly presented, but the method and discussion sections need to be improved prior to publication. Please see my detailed comments below:

Line 34: one point that is not clear in the introduction is why you think study microbial community composition is important? there may be functional redundancy, and different microbes may perform the same or similar function, so function diversity is more important than community composition. –

Reply - Completed in the introduction. We clarified the dependence of microorganisms composition on the quality of their development substrate and the inverse dependence: determination of microorganisms composition shows the quality of their habitat environment

Line 115: what does this table legend mean? need to update the table legend

Reply – the meaning is description of the studied areas and selected samples

Line 118: is it possible to mark sample number in Figure 1 to better help your readers to identify the sample location

Reply - done

Line 143: separate the DNA extraction and sequence method and other physiochemical methods into different paragraphs.

Reply – methods description has been divided on separate sub chapters.

Line 151: need more details about sequence processing, such as Phred score , reads with which length were removed, and adaptor contaminants?

Reply -  All filters were specified in M&Ms. dada2 relies on an algorithm of expected error rate rather than on a simple quality score, witch is used only as a simple primary filter (Q>2).

Line 157: which version of SILVA was used?

Reply – SILVA v. 138

Line 158: would suggest adding another section in the results part to show the sequence depth, before and after quality control sequence reads number, etc.

 Reply  - Indeed, it is reasonable to include primary and final scores in the Results.

Line 228: general comment, be consistent about the text Font and Font size

Reply – the size in changed.

Line 253: did you see any pattern of community composition at genus or family level?

Reply -  Unfortunately no. There is no highly-specific major taxa, connected with specific environmental changes. There were a lot of minor taxa, witch can be described and estimated in more thoughtful work, but in this research we were tried to describe a whole context of microbial changes.

Line 260-263: suggest move this to discussion and can have a more detailed discussion if it's important. The same for other parts, try not to reference previous studies in your results section as you have a separate discussion section.

Reply – corrected, text has been redistributed

Line 366: the discussion section needs to be improved. I did not see any previous studies that were discused against your results. in addition, seems like you observed difference among different sampling locations in community composition, diversity. but the discussion is not enough here to explain why and what does that mean.

Reply – corrected, material has been redistributed

Sincerely yours,

On behalf of the authors group,

Evgeny Abakumov

Reviewer 2 Report

Comments and Suggestions for Authors

much of the interesting discussion is contained within the results for each section and the discussion is therefore somewhat reduced. I understand this was a convenient approach but I felt that some of the key issues about some sites did not come through in the discussion.

Comments on the Quality of English Language

The English is not bad, but does need some grammatical support.

Author Response

Dear reviewer!

Thank you for your comments, point by point replies are given below (tracked in yellow)

Much of the interesting discussion is contained within the results for each section and the discussion is therefore somewhat reduced. I understand this was a convenient approach but I felt that some of the key issues about some sites did not come through in the discussion.

Reply -  We have revised and supplemented the discussion. We hope the new version meets the necessary standards more

Sincerely yours,

On behalf of the authors group,

Evgeny Abakumov

Reviewer 3 Report

Comments and Suggestions for Authors

The work of Dorogaya et al. discusses some properties, particularly the microbiology, of several soil samples collected from mine locations that have ceased operations. The research aims to understand the recovery of soils and ecosystems around mining activities, drawing specifically from the experience of the Republic of Bashkortostan. In my opinion this study can be published pending some minor changes.

Line 50. This should be better explained. How do the specific soil parameters change due to mining? For example, does water infiltration increase or decrease/ In what way do the microbial communities change? Why mining activities change the nutritional composition of soil? Etc. etc.

Line 66. It would be good to better articulate what ae the minerals mined in the Bashkorstan, how many mines are there, etc. Some of this information is reported later in the materials section. I suggest the authors to include a map of the terrain and rebalance the key information for the introduction and materials section and include all redundant information in the supplementary material. A map perhaps could be good too to specify the terrains and put the study in a larger context.

Line 120. It can be specified if these images were a simple google earth search or from personal archives

Line 131. I am not sure if the methods are generally accepted but they are referenced so it is fine

Table 1. “Climatic” is misspelled

Table 2 heading is a typo

Line 136 and following. The methods can just be cited. There is no need to report the entire title in the text

Author Response

Dear reviewer!

Thank you for your comments, point by point replies are given below (tracked in yellow)

The work of Dorogaya et al. discusses some properties, particularly the microbiology, of several soil samples collected from mine locations that have ceased operations. The research aims to understand the recovery of soils and ecosystems around mining activities, drawing specifically from the experience of the Republic of Bashkortostan. In my opinion this study can be published pending some minor changes.

Line 50. This should be better explained. How do the specific soil parameters change due to mining? For example, does water infiltration increase or decrease/ In what way do the microbial communities change? Why mining activities change the nutritional composition of soil? Etc. etc. –

Reply – text has been improved and amended by new information

Line 66. It would be good to better articulate what ae the minerals mined in the Bashkorstan, how many mines are there, etc. Some of this information is reported later in the materials section. I suggest the authors to include a map of the terrain and rebalance the key information for the introduction and materials section and include all redundant information in the supplementary material. A map perhaps could be good too to specify the terrains and put the study in a larger context. –

Reply – text has been improved and amended by new information

Line 120. It can be specified if these images were a simple google earth search or from personal archives

Reply– Fixed: the Google Earth platform was used

Line 131. I am not sure if the methods are generally accepted but they are referenced so it is fine. - With your permission, we will not make corrections to the text. In other works, we will try to avoid this verbal turnover. Thanks for the attentive review!

Reply – the description of methods updated in a current version of the manuscript

Table 1. “Climatic” is misspelled – Fixed

Reply – thank you, mistake is corrected

Table 2 heading is a typo –

Reply – typo has been corrected

Line 136 and following. The methods can just be cited. There is no need to report the entire title in the text

Reply – references on the method are added

Sincerely yours,

On behalf of the authors group,

Evgeny Abakumov

Round 2

Reviewer 1 Report

Comments and Suggestions for Authors

No further comments.